

# What issues are data scientists talking about? Identification of current data science issues using semantic content analysis of Q&A communities

Fatih Gurcan

Department of Management Information Systems, Faculty of Economics and Administrative Sciences, Karadeniz Technical University, Trabzon, Turkey

## ABSTRACT

**Background:** Because of the growing involvement of communities from various disciplines, data science is constantly evolving and gaining popularity. The growing interest in data science-based services and applications presents numerous challenges for their development. Therefore, data scientists frequently turn to various forums, particularly domain-specific Q&A websites, to solve difficulties. These websites evolve into data science knowledge repositories over time. Analysis of such repositories can provide valuable insights into the applications, topics, trends, and challenges of data science.

**Methods:** In this article, we investigated what data scientists are asking by analyzing all posts to date on DSSE, a data science-focused Q&A website. To discover main topics embedded in data science discussions, we used latent Dirichlet allocation (LDA), a probabilistic approach for topic modeling.

**Results:** As a result of this analysis, 18 main topics were identified that demonstrate the current interests and issues in data science. We then examined the topics' popularity and difficulty. In addition, we identified the most commonly used tasks, techniques, and tools in data science. As a result, "Model Training", "Machine Learning", and "Neural Networks" emerged as the most prominent topics. Also, "Data Manipulation", "Coding Errors", and "Tools" were identified as the most viewed (most popular) topics. On the other hand, the most difficult topics were identified as "Time Series", "Computer Vision", and "Recommendation Systems". Our findings have significant implications for many data science stakeholders who are striving to advance data-driven architectures, concepts, tools, and techniques.

# INTRODUCTION

The volume and variety of data produced and shared in today's digital life cycle, also known as the "age of big data", is increasing exponentially on a daily basis. Over the last decade, big data has been at the forefront of the technology-driven revolution in computing ecosystems. Big data applications, including but not limited to search engines, social networks, e-commerce, and multimedia streaming services, have achieved unprecedented success (*Assunção et al., 2015*; *Donoho, 2017*; *Vicario & Coleman, 2019*). As

Corresponding author
Fatih Gurcan, fgurcan@ktu.edu.tr

a result, data-driven services and applications are in high demand and interest across all industrial and social segments (*Cao, 2017*; *Gurcan & Cagiltay, 2019*). This data-driven digital transformation in information technologies has resulted in an increase in activity around data science, and it has thus become an important cornerstone of today's technology-oriented life cycle (*Cao, 2017*; *Sarker, 2021*). Data science is a multidisciplinary field that uses scientific approaches, processes, algorithms, and systems to extract information and insights from structured or unstructured data (*Cao, 2017*; *Sarker, 2021*). Data Science employs a wide range of interdisciplinary techniques and theories from mathematics, statistics, computer science, and information science. Data science fundamentals are based on the combined use of statistical models, data mining, machine learning, and computing methods to analyze and comprehend real-world events using data in various structures (*Cao, 2017*; *Vicario & Coleman, 2019*; *Plotnikova, Dumas & Milani, 2020*; *Sarker, 2021*).

The introduction of big data systems, in particular, has resulted in a significant transformation in the architectures, methodologies, and competencies used in data science (*Donoho, 2017*; *Gurcan, 2019*; *Saltz & Krasteva, 2022*). In other words, data-driven digital transformation has increased the responsibilities of data scientists (*Cao, 2017*). Modern data science applications and practices include a wide range of tasks, technologies, tools, and paradigms. This necessitates data scientists being proficient in a wide range of knowledge-domains and skill sets (*Gurcan, 2019*; *Sarker, 2021*). Even experienced data scientists may struggle to keep up with the rapid evolution of these data-driven paradigms (*Gurcan, 2019*).

For these reasons, data scientists frequently use domain-specific Q&A websites such as data science stack exchange (DSSE) to get help and advice from colleagues about technical challenges (*Stack Exchange, 2022*). DSSE seeks solutions to the problems of data scientists from various backgrounds by managing their thousands of posts (*Stack Exchange, 2022*). With the increase in data science activities in recent years, the DSSE platform has become an important source of information and a useful guide for data scientists (*Karbasian & Johri, 2020*). All of the posts shared on DSSE can be considered as an important data repository that keeps a history of data scientists' topics and thoughts.

Analyzing this repository can provide important insights into which tools and technologies data scientists prefer, the reasons for their choices, the types of work environments and motivations they require, and the technical challenges they face (*Karbasian & Johri, 2020*). From this perspective, researchers have recently conducted many empirical studies using Stack Overflow data to investigate the sub-contexts of software development, such as testing (*Kochhar, 2016*), security (*Yang et al., 2016*), mobile development (*Rosen & Shihab, 2016*), requirements (*Zou et al., 2015*), concurrency development (*Ahmed & Bagherzadeh, 2018*), chatbot development (*Abdellatif et al., 2020*), and IOT development (*Uddin et al., 2021*). Despite these valuable efforts, studies exploring the holistic landscape of data science using semantic content analysis based on topic modeling from online Q&A platforms are still limited, with a few exceptions (*Bagherzadeh & Khatchadourian, 2019*; *Hin, 2020*; *Karbasian & Johri, 2020*). In fact, we believe that

analyzing data from Q&A platforms will have significant implications for understanding the current state of data science.

Taking into account the current background, this study aims to conduct an in-depth and comprehensive analysis of the common issues and challenges faced by data scientists, as well as to fill a gap in the literature. To that end, a semantic content analysis based on probabilistic topic modeling was applied to posts shared on DSSE, a data science-specific Q&A platform, over the last nine years between 2014 and 2022 (*Stack Exchange, 2022*). The semi-automatic methodology proposed in this study includes a series of semantic topic modeling processes based on latent Dirichlet allocation (LDA), a probabilistic topic modeling algorithm used to automatically discover topics (*Blei, Ng & Jordan, 2003*). Using this methodology, we identified the main issues and themes that data scientists are discussing, as well as their underlying dependencies and indicators and long-term trends. In summary, we motivate and present the following research questions for our empirical study.

**RQ1.** What topics are discussed by data scientists?

**RQ2.** How do the data science topics evolve over time?

**RQ3.** How do the popularity and difficulty of the topics vary?

**RQ4.** What are the most commonly used tasks, techniques, and tools in data science?

**RQ5.** How do data science topics relate to data-driven technologies?

## BACKGROUND AND RELATED WORK

To provide a better understanding of the contextual dimensions of data science, we base our research on three pillars and discuss relevant studies here under the headings of data science, Q&A communities, and topic modeling.

### Studies related to the data science

In recent years, the rapid development of data-oriented services and applications has increased the number of studies aimed at understanding the data science paradigms. The majority of domain-specific research focuses on data science concepts, architectures, practices, application areas, big data, and related fields (*Donoho, 2017*). These studies provided insights into the use of data science in academia and industry, as well as data science issues and challenges (*Vicario & Coleman, 2019*; *Karbasian & Johri, 2020*). Previous studies have discussed the use and benefits of data science in various disciplines, ranging from business science to computer sciences, finance to medicine, bioinformatics to natural sciences (*Donoho, 2017*; *Vicario & Coleman, 2019*). In today's data-driven world, common data science application areas include business analytics, supply chain, social media analytics, smart cities, business logistics, business intelligence, recommendation systems, decision support systems, natural language processing, financial fraud detection, advertising, behavioral analytics, and manufacturing (*Schoenherr & Speier-Pero, 2015*; *Vicario & Coleman, 2019*; *Sarker, 2021*). Several studies have emphasized the close relationship of data science with big data, predictive analytics, and data-driven decision making (*Cao, 2017*; *Sarker, 2021*).

A number of studies, similar to our current study, have been conducted based on the analysis of posts from Q&A communities to investigate the issues and challenges of data science (*Bagherzadeh & Khatchadourian, 2019*; *Hin, 2020*; *Karbasian & Johri, 2020*). An empirical study was conducted in which data science-related posts on two Q&A websites, Stack Overflow and Kaggle, were analyzed using the topic modeling approach, and 24 data science-related discussion topics were identified (*Hin, 2020*). In another study, *Karbasian & Johri (2020)* used a topic modeling-based approach to analyze data science-specific posts from two popular Q&A communities, StackExchange and Reddit. The discussion topics discovered on both platforms were compared using their analysis (*Karbasian & Johri, 2020*). Furthermore, *Bagherzadeh & Khatchadourian (2019)* examined Stack Overflow posts to investigate big data-related issues, and as a result, they identified 30 topics. Our study differs from theirs in that we only analyze posts shared on DSSE, a data science-specific Q&A platform, and we propose a semi-automated methodology based on unsupervised machine learning for such analysis. Furthermore, our work complements theirs by adding dimension and depth.

## Studies on Q&A communities

Researchers have long studied online Q&A platforms such as Stack Overflow (*Ahmad et al., 2018*), Stack Exchange (*Karbasian & Johri, 2020*), Quora (*Maity, Sahni & Mukherjee, 2015*), Kaggle (*Hin, 2020*), Yahoo (*Zhang et al., 2020*), and Reddit (*Karbasian & Johri, 2020*). Stack Overflow, in particular, is an important information and experience sharing platform for all developers from various backgrounds under the umbrella of information technologies, and it has been the subject of numerous studies (*Ahmad et al., 2018*).

More specifically, a number of these studies have used Stack Overflow data in order to classify the posts (*Barua, Thomas & Hassan, 2014*; *Beyer et al., 2020*); to investigate descriptive indicators (*i.e.*, tags, view count, score, answer count, *etc.*) of the posts (*Barua, Thomas & Hassan, 2014*; *Yang et al., 2016*; *Uddin et al., 2021*); to analyze the textual contents of posts (*Barua, Thomas & Hassan, 2014*; *Rosen & Shihab, 2016*); to organize its settings, design, and infrastructure (*Treude & Wagner, 2019*; *Alrashedy et al., 2020*); to explore the usability of settings and configurations (*Treude & Wagner, 2019*), and to discover the relationship between questions and their answers (*Xu et al., 2018*).

Stack Overflow data has also been used in a number of studies that revealed discussion topics in specific subfields of software development such as testing (*Kochhar, 2016*), security (*Yang et al., 2016*), mobile development (*Linares-Vásquez, Dit & Poshyvanyk, 2013*; *Rosen & Shihab, 2016*), chatbot development (*Abdellatif et al., 2020*), IOT development (*Uddin et al., 2021*), and machine learning (*Alshangiti et al., 2019*). As previously stated, a small number of studies have also been conducted based on the analysis of posts shared on the Q&A platforms to explore the issues and challenges related to data science (*Bagherzadeh & Khatchadourian, 2019*; *Hin, 2020*; *Karbasian & Johri, 2020*). Our study expands on previous research by analyzing the comprehensive dimensions of posts shared on the DSSE platform, which only includes data science-related posts.

## Topic modeling

Topic modeling is a generative approach for comprehending and summarizing the semantic content of large collections of documents (*Blei, 2012*; *Gurcan et al., 2022b*). It ensures that the word groups that best describe the semantic map of the documents are identified as a separate topic (*Blei, 2012*). Topic modeling is becoming more popular for mining unstructured textual data due to its capabilities and benefits for semantic content analysis (*Gurcan & Cagiltay, 2022*). Moreover, topic modeling is a generative approach actively used to reveal emerging trends in specific contexts of technical sciences such as software engineering, computer sciences, and information sciences (*Silva, Galster & Gilson, 2021*; *Gurcan et al., 2022a*). The amount of data produced in today's software repositories and social networks is growing at an exponential rate. Because of the rapid increase in the amount of unstructured data, semantic content analysis of this data has become difficult but important, even though it provides an opportunity for research on this data (*Silva, Galster & Gilson, 2021*).

From its inception to the present, topic modeling procedures have been successfully applied to various types of data, including textual documents, genetic data, web archives, log files, source codes, images, videos, forums, blogs, Q&A platforms, software repositories, and social networks (*Blei, 2012*; *Silva, Galster & Gilson, 2021*). Furthermore, topic modeling was employed to reveal the major trends in software engineering research (*Mathew & Menzies, 2018*; *Gurcan et al., 2022b*); to explore essential competencies and skills for software engineers by analyzing online jobs (*Gurcan & Kose, 2017*; *Gurcan & Cagiltay, 2019*); to identify what issues are software developers are discussing about specific contexts of software engineering, such as IOT development (*Uddin et al., 2021*), mobile development (*Rosen & Shihab, 2016*), testing (*Kochhar, 2016*), chatbot development (*Abdellatif et al., 2020*) security (*Yang et al., 2016*) and programming languages (*Chakraborty et al., 2021*).

In data science research, the LDA-based topic modeling method was also used to analyze data scientists' discussions on Q&A platforms (*Bagherzadeh & Khatchadourian, 2019*; *Hin, 2020*; *Karbasian & Johri, 2020*). Apart from these aforementioned studies, numerous studies based on topic modeling procedures have been conducted in sub-contexts of various disciplines (*Silva, Galster & Gilson, 2021*). In conclusion, the effectiveness and suitability of the topic model approach for research in the technical sciences has increased our motivation to investigate data science issues using topic models.

# METHOD

## Data collection and extraction

To provide an objective methodology, we used a data dump that included all posts shared on data science stack exchange (DSSE), a data science-specific Q&A platform (*Stack Exchange, 2022*). The datasets created and analyzed during this current study are publicly available in the Internet Archive repository (*Internet Archive, 2022*) as a data dump in XML format. The first step was to download the most recent XML data dump (last updated on June 8, 2022) and parse it into a database. Each question post in the data dump contains

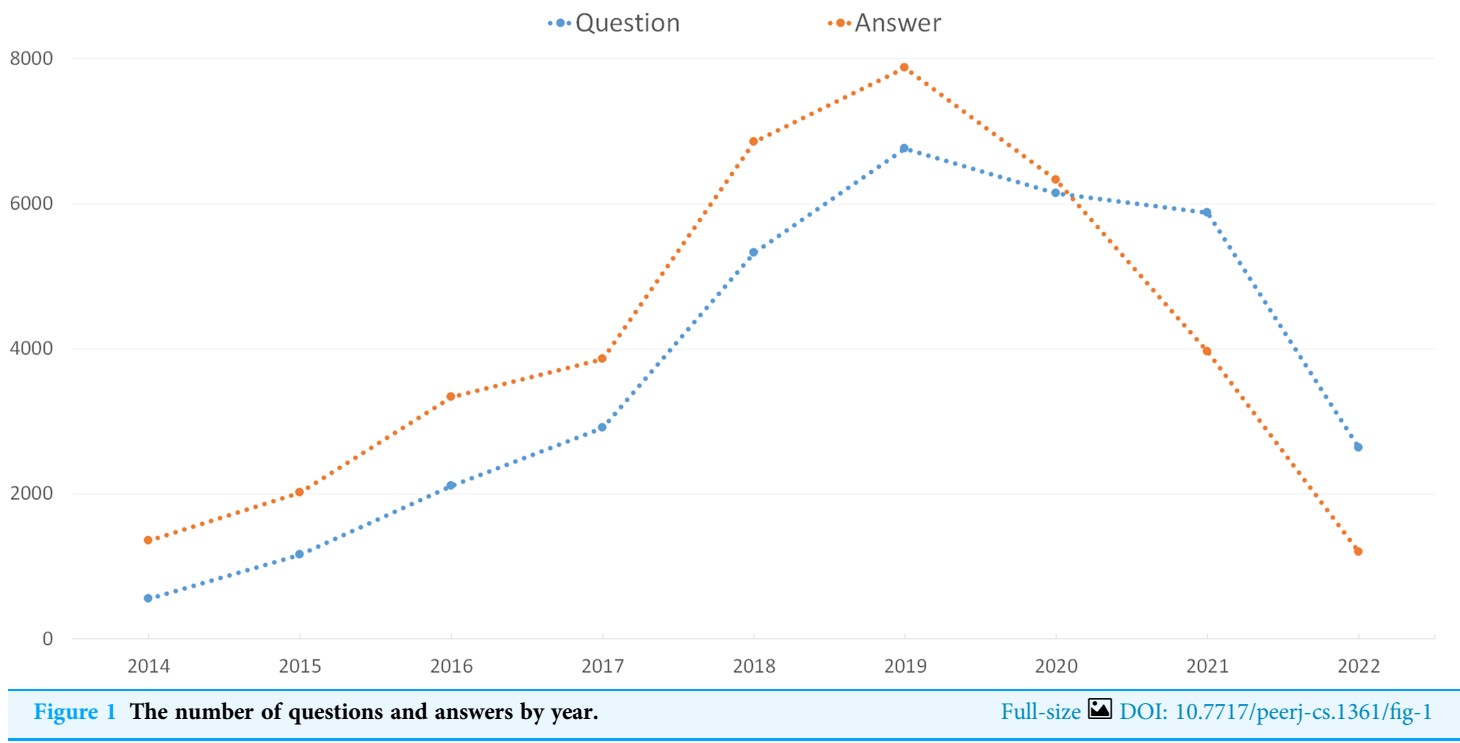

**Figure 1   The number of questions and answers by year.**

various metadata elements such as a title, a body, tags, answers, comments, and other indicators (*Yang et al., 2016*).

From May 2014 to June 2022, this parsed data dump contained 70,283 posts (33,492 questions and 36,791 answers). Figure 1 depicts the number of data science-related posts in our experimental data set by year. According to Fig. 1, the number of questions and answers increased up until 2019 and then decreased. Although the number of answers is generally greater than the number of questions, it has been lower since 2020. It was also discovered that an average of 3,721 questions and 4,088 answers were shared on the DSSE platform each year.

## Data preprocessing

Data preprocessing consists of a series of operations used to convert textual raw data into structured datasets. We considered the preprocessing steps that have been widely preferred in previous work to reduce noise and structure textual data in order to implement an efficient preprocessing (*Rosen & Shihab, 2016*; *Yang et al., 2016*; *Uddin et al., 2021*). Initially, we removed all non-semantic paragraphs and content from a post, such as code snippets marked with <code></code> and non-text blocks marked with HTML tags like <p></p> and <a></a>. In the second step, we removed stop words (*e.g.,* "a", "an", "the", "with", *etc.*), numbers, punctuation, and non-alphabetic characters (*Řehůřek & Sojka, 2011*). In this way, we have removed words and characters from the posts that do not make sense on their own. As a result of this process, we only kept the meaningful words required

for semantic topic modeling in the *corpus*. Following that, we used the lemmatization process to reduce the words to their roots while keeping the various meanings derived from the same root (*Řehůřek & Sojka, 2011*; *Gurcan & Cagiltay, 2022*).

## Implementation of LDA-based topic modeling

Topic modeling is a statistical and computational approach that uses unsupervised machine learning to extract latent semantic patterns from a large collection of documents. For text mining and natural language processing research, several topic modeling algorithms have been proposed, including latent semantic indexing (LSI), latent Dirichlet allocation (LDA), non-negative matrix factorization (NMF), Dirichlet multinomial regression (DMR), hierarchical latent Dirichlet allocation (HLDA), hierarchical Dirichlet process (HDP), dynamic topic model (DTM), and correlated topic model (CTM) (*Řehůřek & Sojka, 2011*; *Gurcan & Cagiltay, 2022*; *Gurcan et al., 2022b*). Unfortunately, the majority of these algorithms do not provide a widely accepted method for calculating the consistency score, which is used to estimate the optimal number of topics (*Gurcan & Cagiltay, 2022*). Among all topic modeling algorithms, LDA provides the fundamental background of topic modeling and is thus more widely used than the others. LDA is widely regarded as one of the most effective techniques for discovering hidden semantic structures known as "topics" in natural language text documents (*Blei, 2012*). In fact, LDA is the most widely used topic modeling algorithm in text mining and natural language processing (*Gurcan et al., 2022a*). In this regard, in this study, we used latent Dirichlet allocation (LDA), a probabilistic model, for topic modeling-based semantic content analysis of a textual *corpus* of data science posts. The theoretical background and graphical model representations of the LDA algorithm is presented in the study by *Blei, Ng & Jordan (2003)*, which introduced the LDA (see Figures 1 and 7 in *Blei, Ng & Jordan (2003)*).

With the aim of implementing the LDA-based topic modeling procedures to our data science *corpus*, we used Gensim (*Řehůřek & Sojka, 2011*), a comprehensive library for text mining and topic modeling. The LDA topic modeling algorithm generates a list of topics from which to group the input preprocessed posts based on the specified K (number of topics). Taking into account previous work (*Uddin et al., 2021*), we used the $C_V$ metric, which is included in the Gensim package (*Řehůřek & Sojka, 2011*), to calculate a coherence score for each K value and to determine the optimal number of K topics for our *corpus*. Consistent with previous work for topic modeling of short texts (*Zuo et al., 2016*; *Gurcan et al., 2022b*), the prior parameters of $\alpha = 0.1$ and $\beta = 0.01$ were used to fine-tune the distribution of topics per document and distribution of words per topic, respectively. The LDA model was implemented using these prior parameters, with K values ranging from 10 to 40. (increasing one at a time). Concurrently, as shown in Fig. 2, we calculated a coherence score ($C_V$) for each topic model implemented for each K (*Řehůřek & Sojka, 2011*). For the number of topics K = 18, the maximum consistency score ($C_V = 0.52589$) was obtained within these $C_V$ metrics, revealing the optimal semantic consistency of the topics. As a result, we determined that 18 was the optimal number of topics.

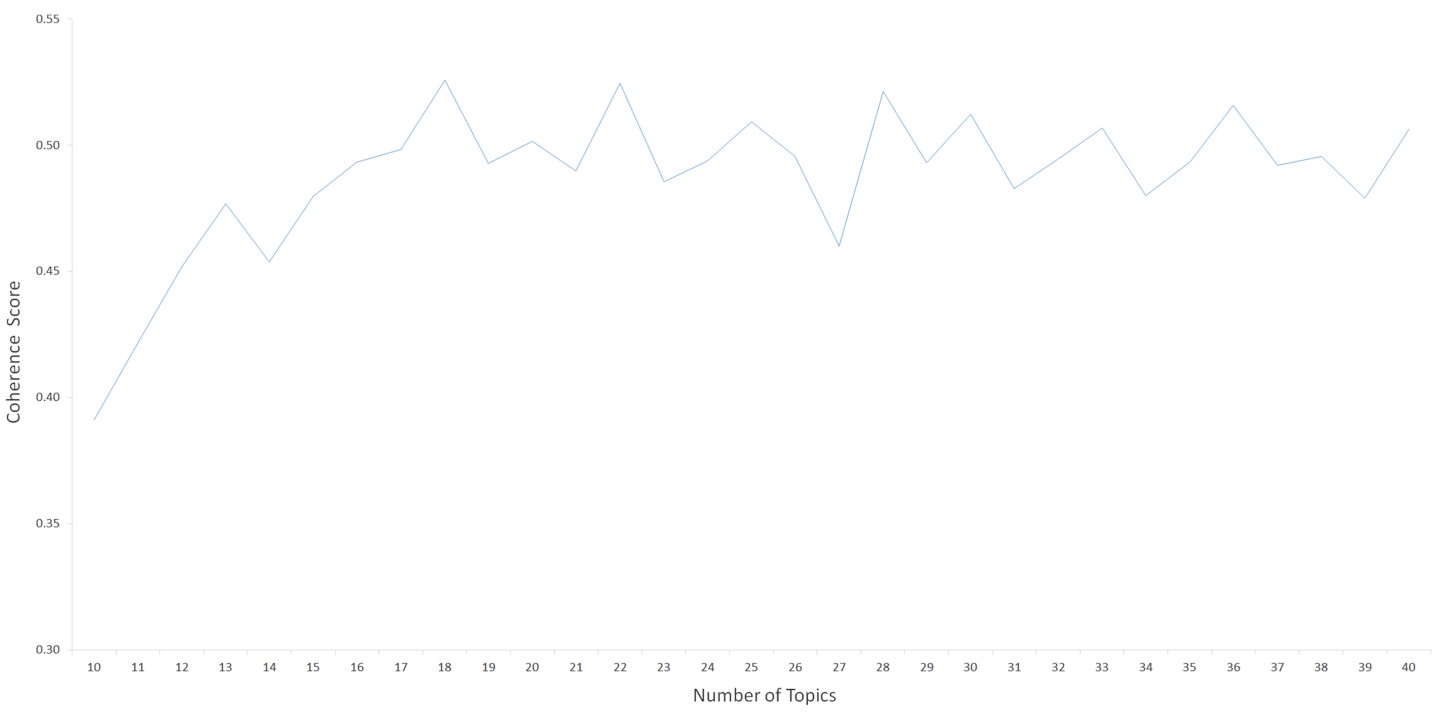

**Figure 2** Coherence scores calculated for each number of topics.

## EMPIRICAL STUDY

In this section, we answer our five research questions:

**RQ1.** What topics are discussed by data scientists?
**RQ2.** How do the data science topics evolve over time?
**RQ3.** How do the popularity and difficulty of the topics vary?
**RQ4.** What are the most commonly used tasks, techniques, and tools in data science?
**RQ5.** How do data science topics relate to data-driven technologies?

### What topics are discussed by data scientists? (RQ1)

*Motivation*

Because of the rapid development of data science paradigms, many innovative architectures, techniques, and tools to support data science development have been developed. To keep up with this rapid change, data scientists must first understand the issues and challenges they face when employing these architectures, techniques, and tools. The number of empirical studies that comprehensively investigate the issues facing data scientists is quite limited. As a result, this type of analysis can aid in understanding the issues that data scientists are discussing.

*Approach*

We used LDA-based topic modeling to investigate the questions that data scientists are asking on the DSSE platform. LDA is a probabilistic and generative topic modeling technique that represents topics as a probability distribution over words in a *corpus*. In the

method section, we describe in detail how we adapted and applied LDA-based topic modeling to our *corpus*. We discovered 18 topics as a result of this LDA analysis, with each topic represented by 30 descriptive keywords. Two independent domain-specific experts assessed the consistency of the topics described with descriptive keywords. Experts evaluated 20 random samples of posts for each of the 18 topics to see if the posts were consistent with the dominant topic to which they were assigned. The topics were then given names based on their keywords. The total percentages of each topic in the entire *corpus* were then calculated, taking into account the dominant topic to which each post is assigned. For example, if a topic has a rate of 10%, it is the dominant topic in 10% of all question posts, and those posts are assigned to that topic.

### Finding

As a result, Table 1 shows the 18 topics discovered by LDA-based topic modeling, along with their topic names, descriptive keywords, and rates. As shown in Table 1, the topics (issues) are listed in descending order by percentage, and the keywords for each topic are also listed in descending order. Because the topics in Table 1 reveal major issues specific to data science, the terms topic and issue are used interchangeably throughout this article. The topics indicated that data scientists face a wide range of issues, from "Machine Learning" to "Model Training", "Neural Networks" to "Time Series", "NLP" to "Computer Vision".

As a result, the top five most frequently asked topics were "Model Training", "Machine Learning", "Neural Networks", "NLP", and "Time Series". The least asked topics, on the other hand, were "Clustering", "Coding Errors" and "Dimensionality Reduction". Based on the issues discovered (see Table 1), we can conclude that machine learning and its subtasks such as "Model Training", "Neural Networks", "Feature Engineering", "Classification", and "Clustering" are dominant topics for data science. Furthermore, the topics of "NLP", "Computer Vision", and "Recommendation Systems" have emerged as the most common application areas of data science.

## How do the data science topics evolve over time? (RQ2)

### Motivation

Data science is a constantly evolving discipline that includes new or outdated tasks, architectures, techniques, and tools. Therefore, the interests, recommendations, and experiences of data scientists may change over time. From this perspective, we attempted to examine how the topics discussed by data scientists have evolved over time. Such an analysis will help to improve understanding of chronic and unsolvable data science issues, as well as generate new solutions to them.

### Approach

At this stage, we examined how data science problems have evolved over time. To achieve this, we looked at the distribution of the number of questions for each topic over time. The annual number of questions for each topic was calculated. Then, for each year, we divided

**Table 1 Details about the topics generated by LDA.**

| Topic name | Descriptive LDA keywords | % |
|---|---|---|
| Model training | Model training train set datum test accuracy dataset validation loss epoch learn split parameter different performance increase prediction run rate cross improve time best hyperparameter | 8.88 |
| Machine learning | Learn datum machine algorithm problem learning project deep example ML approach task want method best analysis technique research model lot book idea give field kind | 7.81 |
| Neural networks | Network layer neural output input weight number size LSTM activation neuron convolutional function learn architecture CNN hide keras batch example sequence train convolution deep NN | 7.76 |
| NLP | Word text document sentence model example topic want NLP extract embedding language name give tag entity sentiment BERT token way embed similar task approach *corpus* | 7.20 |
| Time series | Time datum series predict model day prediction LSTM value sequence event problem month forecast anomaly want point different example dataset future window time-series give input | 6.04 |
| Data manipulation | Column datum row value dataframe want dataset file name table number list create code way contain panda ID python string CSV convert format frame different example date output set multiple | 6.03 |
| Feature engineering | Feature variable value datum categorical dataset target correlation encode scale numerical continuous model column problem category way predict transform apply different binary independent want encoding | 5.91 |
| Regression models | Regression feature model tree linear random decision forest logistic XGBoost algorithm importance prediction parameter method selection predict fit boost coefficient variable best sklearn different classifier | 5.86 |
| Statistics | Datum distribution sample number point mean value probability set way variance standard outlier example test group parameter give case different estimate random method approach size | 5.48 |
| Tools | Python run file datum code R package memory library GPU time google notebook spark script tool tensorflow way database version want create load process orange | 5.39 |
| Computer vision | Image object model CNN dataset detection train box want detect pixel different video color segmentation picture example recognition size label problem map face classify extract | 5.35 |
| Classification | Class classification label dataset classifier problem datum binary model sample predict classify probability imbalance set example train training prediction positive give category accuracy case Bayes | 4.98 |
| Recommendation systems | User product customer item datum base system predict want problem model dataset sale recommendation category number rating approach give price buy example build purchase group | 4.81 |
| Loss functions | Function loss gradient weight amp value equation cost error descent update give calculate formula problem mean derivative optimization parameter compute square sum entropy term define | 4.55 |
| Evaluation | Score plot value metric negative curve precision positive calculate matrix mean graph show recall line average threshold confusion code true compare evaluate evaluation interpret performance | 3.63 |
| Clustering | Cluster algorithm action state graph point clustering reward node distance K-means game agent reinforcement problem policy player learn value give method number take different space | 3.57 |
| Coding errors | Error code array shape model keras tensorflow input return function import tensor run custom NumPy line output value error fault call expect problem message pass create | 3.54 |
| Dimensionality reduction | Vector matrix dimension feature similarity distance PCA space autoencoder kernel point representation datum attention component dimensionality embed example cosine represent encoder reduction reduce dimensional calculate | 3.21 |

the number of questions per topic by the total number of questions for that year. In this way, we normalized the question distribution of each topic as a percentage for that year. In other words, we calculated the percentage of questions for each topic per year. Then we subtract the previous year's percentages from the current year's percentages. As a result, we calculated how much the topics changed in the current year compared to the previous year. Finally, we calculated the overall temporal trend of the topics by adding the annual percentage changes for each topic.

**Table 2 Temporal trends of the topics.**

| Topic name | Total rate | Yearly rates (%) | | | | | | | | | Trend value | Trend |
|---|---|---|---|---|---|---|---|---|---|---|---|---|
| | | 2014 | 2015 | 2016 | 2017 | 2018 | 2019 | 2020 | 2021 | 2022 | | |
| Model training | 8.88 | 2.33 | 3.45 | 5.44 | 6.41 | 8.26 | 9.74 | 10.56 | 10.39 | 9.87 | 7.54 | ⇑ |
| Machine learning | 7.81 | 19.50 | 17.83 | 11.88 | 9.70 | 7.72 | 7.07 | 6.09 | 5.87 | 5.96 | −13.54 | ⇓ |
| Neural networks | 7.76 | 1.79 | 3.88 | 7.90 | 11.48 | 9.74 | 7.89 | 7.10 | 6.86 | 5.69 | 3.90 | ⇑ |
| NLP | 7.20 | 8.41 | 7.84 | 7.71 | 6.92 | 5.91 | 6.54 | 7.97 | 7.81 | 7.67 | −0.74 | ⇓ |
| Time series | 6.04 | 5.19 | 3.10 | 5.35 | 5.01 | 5.95 | 5.99 | 5.87 | 6.62 | 8.61 | 3.43 | ⇑ |
| Data manipulation | 6.03 | 4.29 | 6.63 | 6.06 | 5.25 | 6.46 | 6.69 | 5.65 | 6.07 | 5.24 | 0.94 | ⇑ |
| Feature engineering | 5.91 | 4.83 | 5.60 | 4.64 | 5.25 | 5.84 | 5.79 | 6.51 | 6.14 | 6.57 | 1.74 | ⇑ |
| Regression models | 5.86 | 3.40 | 6.03 | 6.34 | 5.90 | 5.52 | 5.65 | 6.15 | 5.61 | 7.02 | 3.62 | ⇑ |
| Statistics | 5.48 | 5.37 | 5.94 | 6.34 | 5.59 | 5.37 | 4.71 | 6.00 | 5.58 | 5.24 | −0.13 | ⇓ |
| Tools | 5.39 | 17.89 | 9.91 | 7.57 | 6.10 | 5.48 | 5.12 | 4.25 | 4.19 | 4.06 | −13.83 | ⇓ |
| Computer vision | 5.35 | 1.97 | 1.89 | 2.74 | 4.77 | 5.90 | 5.55 | 5.61 | 6.57 | 5.35 | 3.38 | ⇑ |
| Classification | 4.98 | 5.01 | 4.48 | 3.79 | 5.07 | 5.09 | 5.00 | 4.91 | 5.36 | 5.09 | 0.08 | ⇑ |
| Recommendation systems | 4.81 | 8.23 | 6.55 | 7.62 | 5.62 | 4.32 | 4.01 | 4.48 | 4.47 | 4.74 | −3.49 | ⇓ |
| Loss functions | 4.55 | 2.15 | 3.79 | 3.74 | 4.77 | 4.79 | 5.12 | 4.96 | 4.03 | 4.10 | 1.95 | ⇑ |
| Evaluation | 3.63 | 1.25 | 3.19 | 3.41 | 3.12 | 3.00 | 3.55 | 4.51 | 3.78 | 4.21 | 2.96 | ⇑ |
| Clustering | 3.57 | 3.94 | 4.82 | 4.40 | 3.81 | 4.07 | 3.43 | 2.93 | 3.37 | 3.34 | −0.60 | ⇓ |
| Coding errors | 3.54 | 0.72 | 0.78 | 1.61 | 2.26 | 3.59 | 4.84 | 3.60 | 3.98 | 3.80 | 3.08 | ⇑ |
| Dimensionality reduction | 3.21 | 3.76 | 4.31 | 3.45 | 2.98 | 2.99 | 3.31 | 2.85 | 3.30 | 3.45 | −0.30 | ⇓ |

*Finding*

Table 2 shows the annual percentages, total trends, and trend directions for the topics, which are listed in descending order by overall percentage. Table 2 shows the annual percentage changes for each topic in each row, and a number of inferences can be drawn about how the topics have evolved over time. Figure 3 depicts the total trend values of the topics in descending order to provide a broader understanding. Figure 3 demonstrates 11 topics with an increasing trend and seven topics with a decreasing trend. As seen in Fig. 3, "Model Training", "Neural Networks", "Regression Models", "Time Series", and "Computer Vision" are the top five topics with the most increasing trend, while "Tools", "Machine Learning", The topics "Recommendation Systems", "NLP" and "Clustering" have the most decreasing trend.

## How do the popularity and difficulty of the topics vary? (RQ3)

*Motivation*

Our findings at RQ1 revealed the broad scope and diversity of data science issues. Data scientists discuss various technical issues at various levels in order to understand specific tasks, techniques, and tools and how they operate effectively. As a result, these issues are unlikely to have the same level of popularity and difficulty. Some questions are displayed numerous times, while others are not displayed at all. Identifying the popularity and difficulty of data science issues can help prioritize efforts to improve data science. Taking

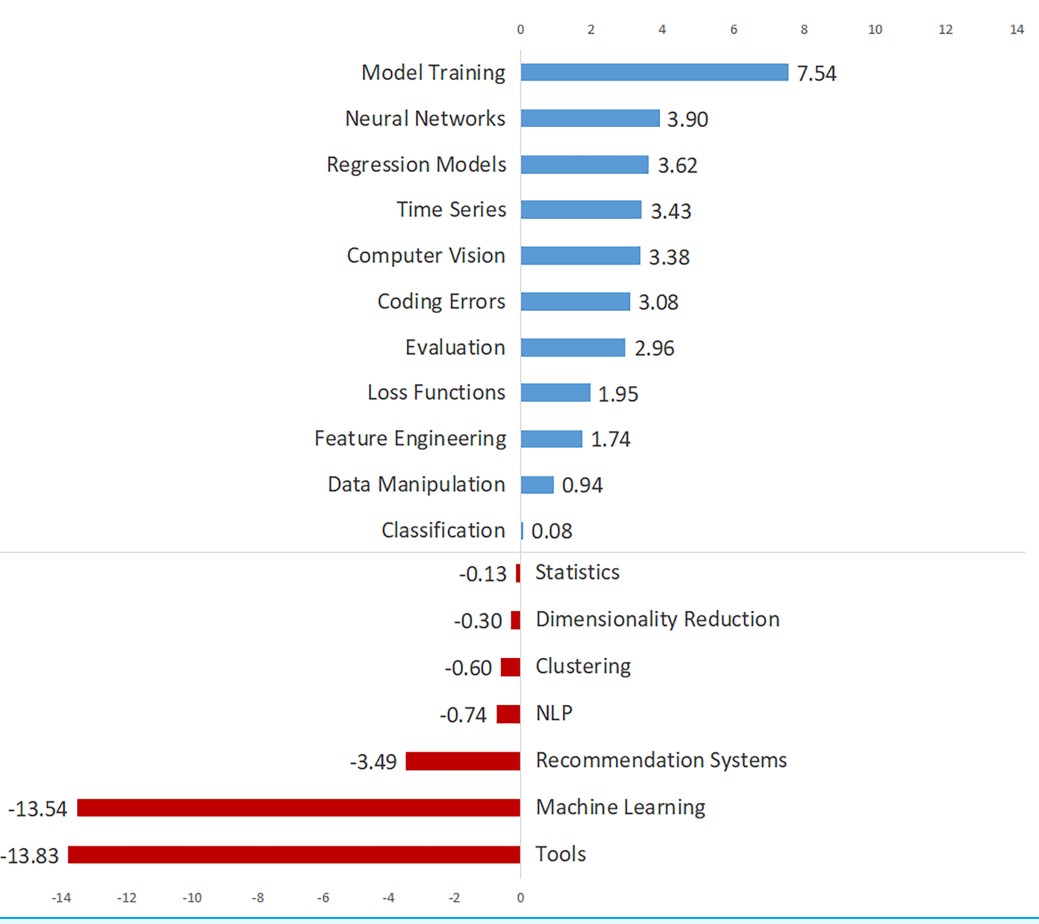

**Figure 3** The topics with increasing and decreasing trends.

into account the implications of this analysis, for example, data science beginners can concentrate on more popular topics. Tool builders can create tools that provide solutions to more difficult data science problems.

### Approach

A question post on DSSE includes descriptive indicators such as the number of views, answers, accepted answers, score, favorites, and comments. After a user posts a question on DSSE, other users respond with answers. When a user realizes an answer that resolves their question, that user marks that answer as the accepted answer for that question. In this way, the user indicates that the problem has been resolved, and other users who encounter the same issue can view this solution. At this stage, we performed a series of computational analyses using these indicators to determine the difficulty and popularity of each topic. We began by calculating the number of questions assigned to each topic. Next, we divided the total number of views for each topic by the total number of questions for each topic to calculate the average number of views for each topic. We calculated the popularity of the topics in this way. We then calculated the average of the number of accepted answers for each topic by dividing the number of accepted answers for that topic by the number of questions for that topic. We determined the difficulty of the topics in this manner.

**Table 3 Descriptive indicators of the data science topics.**

| Topic name | Question count (%) | Question count (#) | View count (#) | Answer count (#) | Comment count (#) | Favorite count (#) | Score (#) | Accepted answer count (%) |
|---|---|---|---|---|---|---|---|---|
| Model training | 8.88 | 2,973 | 1,656 | 1.14 | 1.23 | 0.57 | 1.89 | 33.10 |
| Machine learning | 7.81 | 2,615 | 1,485 | 1.40 | 1.24 | 1.22 | 2.84 | 33.19 |
| Neural networks | 7.76 | 2,598 | 2,060 | 1.04 | 0.93 | 1.02 | 2.72 | 37.34 |
| NLP | 7.20 | 2,411 | 1,211 | 1.07 | 0.96 | 0.55 | 1.84 | 29.45 |
| Time series | 6.04 | 2,023 | 828 | 0.90 | 1.10 | 0.52 | 1.51 | 20.56 |
| Data manipulation | 6.03 | 2,020 | 4,491 | 1.23 | 1.05 | 0.40 | 1.61 | 41.39 |
| Feature engineering | 5.91 | 1,979 | 2,007 | 1.23 | 1.10 | 0.88 | 2.15 | 35.17 |
| Regression models | 5.86 | 1,964 | 2,079 | 1.16 | 1.00 | 0.76 | 2.29 | 38.19 |
| Statistics | 5.48 | 1,835 | 983 | 1.02 | 1.15 | 0.53 | 1.79 | 29.92 |
| Tools | 5.39 | 1,805 | 2,308 | 1.15 | 1.06 | 0.73 | 2.48 | 32.02 |
| Computer vision | 5.35 | 1,791 | 911 | 0.87 | 0.96 | 0.42 | 1.57 | 25.52 |
| Classification | 4.98 | 1,668 | 1,923 | 1.16 | 1.22 | 0.75 | 2.29 | 33.75 |
| Recommendation systems | 4.81 | 1,611 | 554 | 1.06 | 1.02 | 0.37 | 1.37 | 27.25 |
| Loss functions | 4.55 | 1,525 | 1,538 | 1.01 | 0.87 | 0.86 | 2.44 | 38.30 |
| Evaluation | 3.63 | 1,217 | 1,922 | 1.09 | 1.06 | 0.48 | 1.81 | 36.15 |
| Clustering | 3.57 | 1,197 | 1,245 | 0.99 | 1.14 | 0.57 | 1.93 | 32.25 |
| Coding errors | 3.54 | 1,186 | 3,771 | 0.96 | 1.03 | 0.33 | 1.38 | 32.55 |
| Dimensionality reduction | 3.21 | 1,074 | 1,299 | 1.04 | 0.80 | 0.81 | 2.44 | 33.05 |

Similarly, we calculated the average number of answers, favorites, comments, and score for each topic. Since the average number of accepted answers ranges from 0 to 1, we presented this indicator as a percentage.

### Finding

Table 3 shows the calculated averages of the questions, views, answers, scores, comments, accepted answers, and favorites for each topic in order to highlight all dimensions of data scientists' interest in domain-specific issues. The topics in this table are listed in descending order of their percentages. The findings in Table 3 provide insight into the interests and perspectives of data scientists on various issues. Moreover, we calculated the popularity of the topics based on the average number of views and presented it in Fig. 4. As shown in Fig. 4, the top five most viewed (most popular) topics are "Data Manipulation", "Coding Errors", "Tools", "Regression Models", and "Neural Networks". On the other hand, the least viewed (least popular) topics are "Recommendation Systems", "Time Series", and "Computer Vision". Also, we determined the difficulty of the topics by considering the average number of accepted answers and presented them in Fig. 5. According to the difficulty metrics of the topics shown in Fig. 5, the most difficult topic is "Time Series", which has the lowest rate (21%) based on the number of accepted answers. The other most

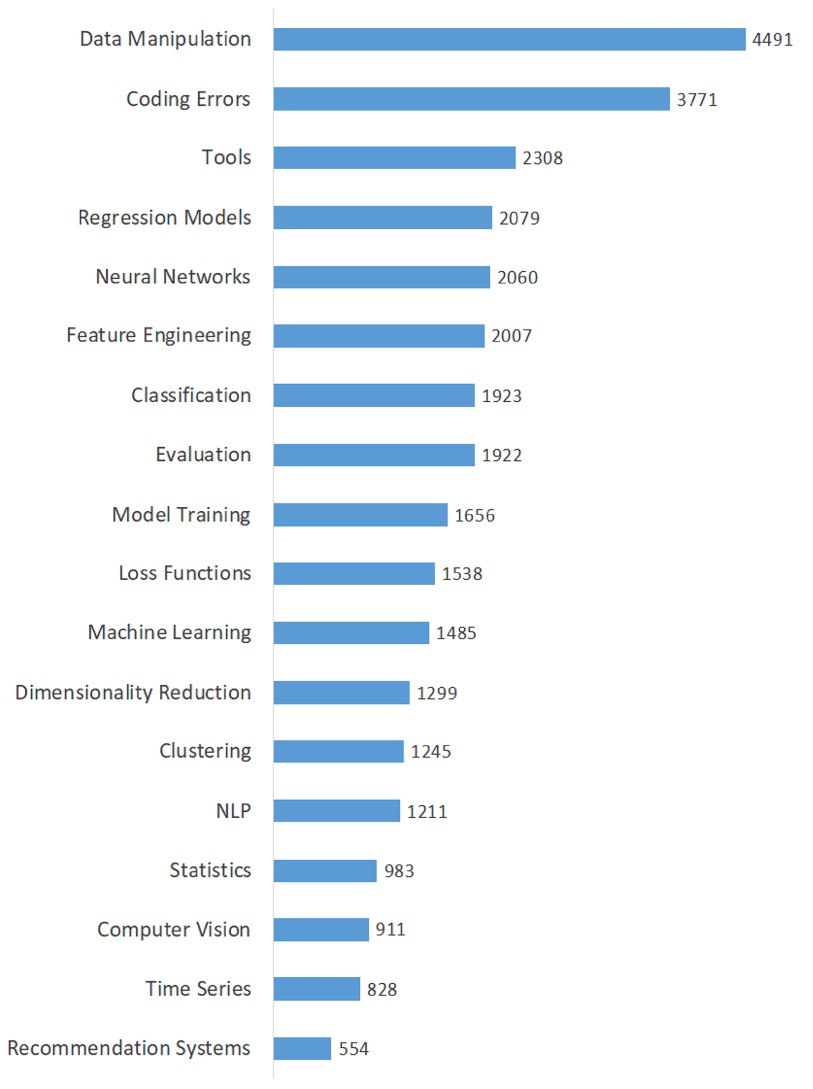

**Figure 4  Average number of views of the topics.**     

difficult topics are "Computer Vision", "Recommendation Systems", "NLP", and "Statistics".

## What are the most commonly used tasks, techniques, and tools in data science? (RQ4)

### Motivation

The growing interest in data-driven applications and services has resulted in an expansion and diversification of data science technologies. Innovative data technologies, which cover a wide range of tasks, methods, and tools, are now widely used in today's data science environments. As a result, the majority of data science issues and challenges are closely related to these tasks, techniques, and tools. It is highly possible that trends in data science technologies will evolve in concurrently with the technological transformations experienced in data-driven ecosystems. Identifying the most commonly used tasks,

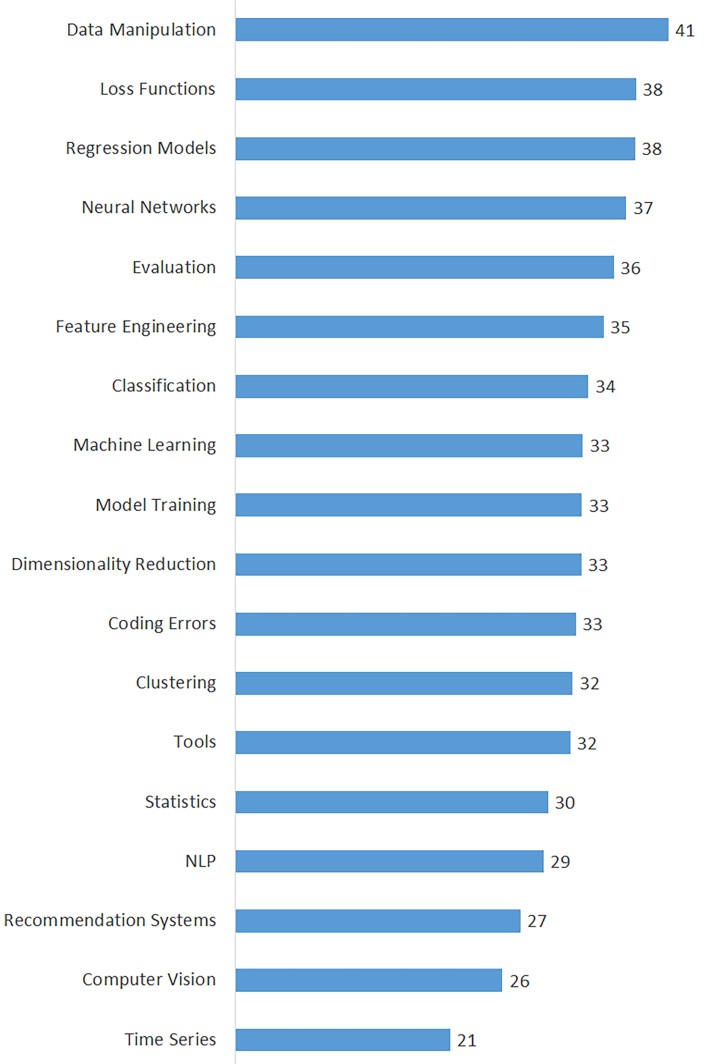

**Figure 5  Percentage of accepted answers of the topics.**

techniques, and tools in data science can provide important insights into various dimensions of data science issues. An analysis of this nature will also help in understanding the relative popularity of data-driven technologies over time. Revealing useful and popular data technologies can help data scientists choose the appropriate tools to advance data science.

## Approach

Each DSSE question post contains tags that provide context and background for that question. These tags are chosen and added to the question by the user who asked it. The tags are descriptive keywords that display data science-related themes, tasks, techniques, and tools that users associate with their questions. To extract a definitive set of all tags used in data science, we first separated the tags of each post into individual tags and calculated the tag frequencies across all posts. Following that, we identified the tags with the highest

**Table 4 Time-based trends in the top 50 data-driven technologies.**

| Tags | Total rate | Yearly rates (%) | | | | | | | | | Trend value | Trend |
|---|---|---|---|---|---|---|---|---|---|---|---|---|
| | | 2014 | 2015 | 2016 | 2017 | 2018 | 2019 | 2020 | 2021 | 2022 | | |
| Machine-learning | 12.28 | 15.63 | 16.61 | 13.19 | 13.77 | 13.08 | 12.33 | 11.53 | 10.80 | 11.13 | −4.50 | ⇓ |
| Python | 7.28 | 3.07 | 4.91 | 6.15 | 6.56 | 7.43 | 8.34 | 7.36 | 7.51 | 6.66 | 3.59 | ⇑ |
| Deep-learning | 5.28 | 0.65 | 1.22 | 3.08 | 5.78 | 6.30 | 5.18 | 5.14 | 5.79 | 5.96 | 5.31 | ⇑ |
| Neural-network | 4.85 | 2.33 | 3.98 | 5.53 | 6.66 | 6.60 | 4.75 | 4.30 | 3.65 | 3.54 | 1.21 | ⇑ |
| Classification | 3.57 | 6.79 | 5.58 | 4.04 | 3.88 | 3.41 | 3.35 | 3.31 | 3.27 | 3.78 | −3.01 | ⇓ |
| Keras | 3.11 | 0.00 | 0.08 | 1.19 | 2.37 | 3.76 | 4.10 | 3.36 | 3.12 | 2.37 | 2.37 | ⇑ |
| NLP | 2.78 | 3.63 | 2.31 | 2.35 | 2.37 | 2.10 | 2.57 | 3.03 | 3.50 | 3.37 | −0.26 | ⇓ |
| Scikit-learn | 2.48 | 1.30 | 1.76 | 2.85 | 2.47 | 2.50 | 2.58 | 2.68 | 2.36 | 2.11 | 0.81 | ⇑ |
| TensorFlow | 2.46 | 0.00 | 0.17 | 1.37 | 2.36 | 2.57 | 2.39 | 2.76 | 3.07 | 2.50 | 2.50 | ⇑ |
| Time-series | 1.97 | 2.14 | 1.09 | 1.93 | 1.76 | 1.68 | 1.92 | 1.76 | 2.38 | 2.92 | 0.78 | ⇑ |
| Regression | 1.68 | 1.40 | 1.80 | 1.63 | 1.90 | 1.58 | 1.56 | 1.89 | 1.49 | 1.94 | 0.55 | ⇑ |
| R | 1.67 | 4.93 | 6.38 | 4.20 | 2.79 | 1.42 | 1.11 | 1.15 | 0.95 | 1.21 | −3.72 | ⇓ |
| Dataset | 1.60 | 4.00 | 2.85 | 1.89 | 1.72 | 1.39 | 1.43 | 1.49 | 1.67 | 1.33 | −2.67 | ⇓ |
| CNN | 1.53 | 0.00 | 0.00 | 0.02 | 0.43 | 1.90 | 2.19 | 1.73 | 1.70 | 1.30 | 1.30 | ⇑ |
| Clustering | 1.53 | 3.44 | 3.10 | 2.59 | 1.79 | 1.43 | 1.33 | 1.29 | 1.30 | 1.42 | −2.02 | ⇓ |
| Pandas | 1.41 | 0.74 | 0.34 | 1.01 | 1.16 | 1.44 | 1.81 | 1.48 | 1.41 | 1.24 | 0.50 | ⇑ |
| Data-mining | 1.34 | 6.79 | 5.62 | 3.06 | 2.19 | 1.25 | 1.00 | 0.77 | 0.65 | 0.66 | −6.13 | ⇓ |
| LSTM | 1.32 | 0.00 | 0.00 | 0.02 | 0.55 | 1.70 | 1.67 | 1.44 | 1.26 | 1.90 | 1.90 | ⇑ |
| Predictive-modeling | 1.31 | 1.58 | 2.98 | 3.00 | 1.59 | 1.13 | 1.22 | 1.08 | 0.92 | 1.04 | −0.54 | ⇓ |
| Statistics | 1.21 | 3.63 | 2.52 | 1.45 | 1.08 | 0.94 | 0.99 | 1.30 | 1.22 | 1.27 | −2.36 | ⇓ |
| Feature-selection | 1.05 | 1.77 | 1.01 | 1.27 | 1.19 | 0.90 | 1.01 | 1.05 | 1.08 | 0.90 | −0.86 | ⇓ |
| Random-forest | 0.93 | 0.93 | 1.59 | 1.05 | 1.11 | 0.78 | 0.77 | 1.07 | 0.87 | 0.87 | −0.06 | ⇓ |
| Data | 0.90 | 0.00 | 0.96 | 1.13 | 0.94 | 0.72 | 0.93 | 0.89 | 0.94 | 1.00 | 1.00 | ⇑ |
| Image-classification | 0.85 | 0.00 | 0.71 | 0.80 | 0.67 | 0.91 | 0.93 | 0.85 | 0.95 | 0.70 | 0.70 | ⇑ |
| RNN | 0.84 | 0.00 | 0.17 | 0.60 | 0.89 | 1.35 | 0.94 | 0.78 | 0.57 | 0.80 | 0.80 | ⇑ |
| Machine-learning-model | 0.82 | 0.00 | 0.00 | 0.02 | 0.00 | 0.68 | 1.00 | 0.96 | 1.20 | 1.47 | 1.47 | ⇑ |
| Decision-trees | 0.82 | 0.09 | 1.01 | 1.00 | 0.93 | 0.91 | 0.68 | 0.89 | 0.82 | 0.63 | 0.53 | ⇑ |
| Data-cleaning | 0.81 | 1.21 | 1.30 | 1.27 | 0.78 | 0.76 | 0.68 | 0.77 | 0.85 | 0.66 | −0.55 | ⇓ |
| Linear-regression | 0.80 | 0.00 | 0.96 | 0.96 | 0.58 | 0.90 | 0.82 | 0.85 | 0.66 | 0.92 | 0.92 | ⇑ |
| Convolutional-neural-network | 0.79 | 0.00 | 0.08 | 1.11 | 1.50 | 0.96 | 0.50 | 0.32 | 1.18 | 0.84 | 0.84 | ⇑ |
| Logistic-regression | 0.75 | 1.40 | 1.38 | 0.84 | 0.83 | 0.60 | 0.70 | 0.73 | 0.70 | 0.92 | −0.48 | ⇓ |
| XGBoost | 0.73 | 0.00 | 0.29 | 0.86 | 1.04 | 0.49 | 0.79 | 0.89 | 0.65 | 0.70 | 0.70 | ⇑ |
| Training | 0.71 | 0.00 | 0.00 | 0.38 | 0.64 | 0.68 | 0.63 | 0.92 | 0.86 | 0.86 | 0.86 | ⇑ |
| Visualization | 0.69 | 1.77 | 1.22 | 1.41 | 1.00 | 0.54 | 0.60 | 0.58 | 0.62 | 0.43 | −1.34 | ⇓ |
| Cross-validation | 0.68 | 0.47 | 0.34 | 0.76 | 0.55 | 0.70 | 0.72 | 0.86 | 0.59 | 0.57 | 0.10 | ⇑ |
| Reinforcement-learning | 0.68 | 0.19 | 0.38 | 0.38 | 0.64 | 0.90 | 0.83 | 0.53 | 0.66 | 0.70 | 0.52 | ⇑ |
| Feature-engineering | 0.68 | 0.00 | 0.21 | 0.58 | 0.65 | 0.56 | 0.76 | 0.73 | 0.80 | 0.73 | 0.73 | ⇑ |
| Pytorch | 0.68 | 0.00 | 0.00 | 0.00 | 0.18 | 0.26 | 0.69 | 0.81 | 1.35 | 1.15 | 1.15 | ⇑ |
| Computer-vision | 0.67 | 0.09 | 0.29 | 0.40 | 0.43 | 0.85 | 0.52 | 0.69 | 0.83 | 1.01 | 0.92 | ⇑ |
| Text-mining | 0.65 | 3.35 | 1.93 | 1.43 | 1.22 | 0.67 | 0.45 | 0.40 | 0.32 | 0.28 | −3.07 | ⇓ |

| Table 4 (continued) | | | | | | | | | | | | |
|---|---|---|---|---|---|---|---|---|---|---|---|---|
| **Tags** | **Total rate** | **Yearly rates (%)** | | | | | | | | | **Trend value** | **Trend** |
| | | **2014** | **2015** | **2016** | **2017** | **2018** | **2019** | **2020** | **2021** | **2022** | | |
| Data-science-model | 0.63 | 0.00 | 0.00 | 0.00 | 0.01 | 0.44 | 0.85 | 0.77 | 0.87 | 1.03 | 1.03 | ⇑ |
| SVM | 0.63 | 1.40 | 1.47 | 0.68 | 1.11 | 0.52 | 0.57 | 0.59 | 0.46 | 0.43 | −0.97 | ⇓ |
| Loss-function | 0.59 | 0.00 | 0.08 | 0.28 | 0.29 | 0.52 | 0.70 | 0.69 | 0.73 | 0.72 | 0.72 | ⇑ |
| Class-imbalance | 0.58 | 0.47 | 0.38 | 0.36 | 0.53 | 0.54 | 0.61 | 0.67 | 0.72 | 0.44 | −0.02 | ⇓ |
| Multiclass-classification | 0.58 | 0.00 | 0.76 | 0.46 | 0.57 | 0.55 | 0.59 | 0.56 | 0.63 | 0.72 | 0.72 | ⇑ |
| Preprocessing | 0.56 | 0.00 | 0.59 | 0.44 | 0.50 | 0.52 | 0.55 | 0.57 | 0.64 | 0.70 | 0.70 | ⇑ |
| Optimization | 0.52 | 1.02 | 0.71 | 0.62 | 0.42 | 0.43 | 0.50 | 0.50 | 0.57 | 0.61 | −0.41 | ⇓ |
| Word-embeddings | 0.52 | 0.00 | 0.42 | 0.48 | 0.37 | 0.48 | 0.56 | 0.56 | 0.53 | 0.69 | 0.69 | ⇑ |
| Recommender-system | 0.51 | 2.05 | 0.96 | 0.92 | 0.57 | 0.42 | 0.37 | 0.44 | 0.46 | 0.58 | −1.46 | ⇓ |
| Unsupervised-learning | 0.51 | 0.00 | 0.29 | 0.64 | 0.60 | 0.49 | 0.48 | 0.55 | 0.50 | 0.46 | 0.46 | ⇓ |

frequency across the entire *corpus*. Then, taking into account their annual frequency distributions, we calculated the annual percentages of these tags. We calculated how each tag changed in that year compared to the previous year by subtracting the percentage of each tag in the previous year from the percentage in the current year. In this way, we determined the percentage increase or decrease of tag frequencies for each year. Following that, we calculated the overall trend for each tag over time by adding these annual changes. Finally, we identified the most commonly used tasks, techniques, and tools in data science by categorizing the tags according to their functions and contexts.

### Finding

We identified 665 unique tags used in data science by analyzing the tags of all posts in the *corpus*. We calculated the annual percentages and total percentages of the top 50 tags with the highest frequency and presented them in Table 4 in descending order of their percentages to provide a clearer understanding of the tags. Table 4 demonstrates the top 50 tags' key findings, including yearly rates (from 2014 to 2022), total rates, trend values, and trend directions. Figure 6 also shows the top 20 tags with the highest frequency and their percentages. According to Fig. 6, "machine-learning" is the most prominent tag, followed by "python", "deep learning", "neural network", and "classification". Consistent with the data science topics discovered by the LDA (see Table 1), machine learning-related tags such as "machine learning", "deep learning", and "neural network" emerged as clearly dominant.

In order to demonstrate the temporal trends of these 50 tags, we also present the annual percentages, trend values, and trend directions of the tags in Table 4. In Fig. 7, we visualized the top 15 tags with increasing and decreasing trends to better illustrate the tag trends. As seen in Fig. 7, the prominent tags with an increasing trend are "deep-learning", "python", "tensorflow", "keras" and "lstm". On the other hand, top tags with decreasing trend are identified as "data-mining", "machine-learning", "r", "text-mining" and "classification". Considering these trends, we can predict that deep learning will become a

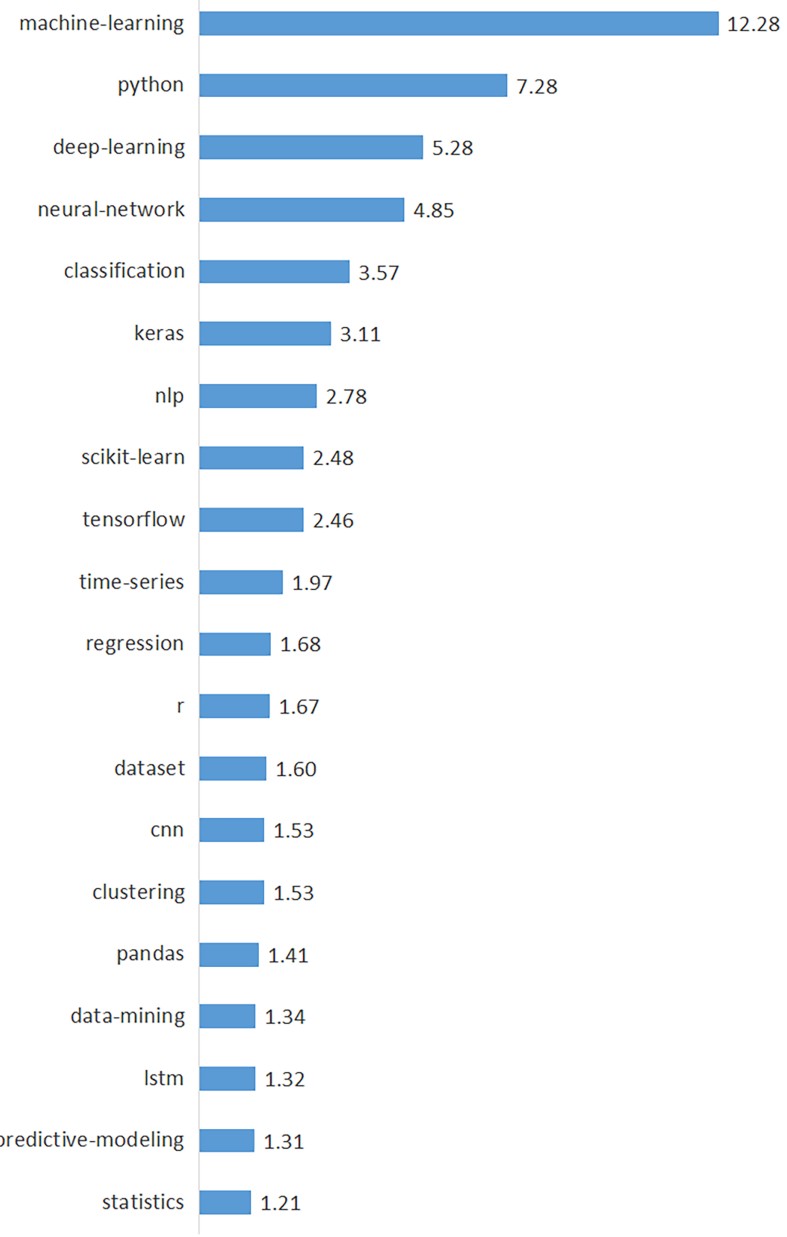

**Figure 6** **Percentages of top 20 data-driven technologies.**

widespread research and application area for data scientists in the near future. It is an interesting finding that, while the trend of "data-mining" and "machine-learning" tags is decreasing, the trend of "deep-learning" and related tags such as "tensorflow", "keras", and "lstm" is increasing. This finding clearly shows a shift in data science from "machine-learning" to "deep-learning".

The topics and tags discovered in the previous stage of our analysis revealed that data science issues cover a wide range of data science tasks, techniques, and tools. When the context and background of data science issues are thoroughly examined, it is clear that the
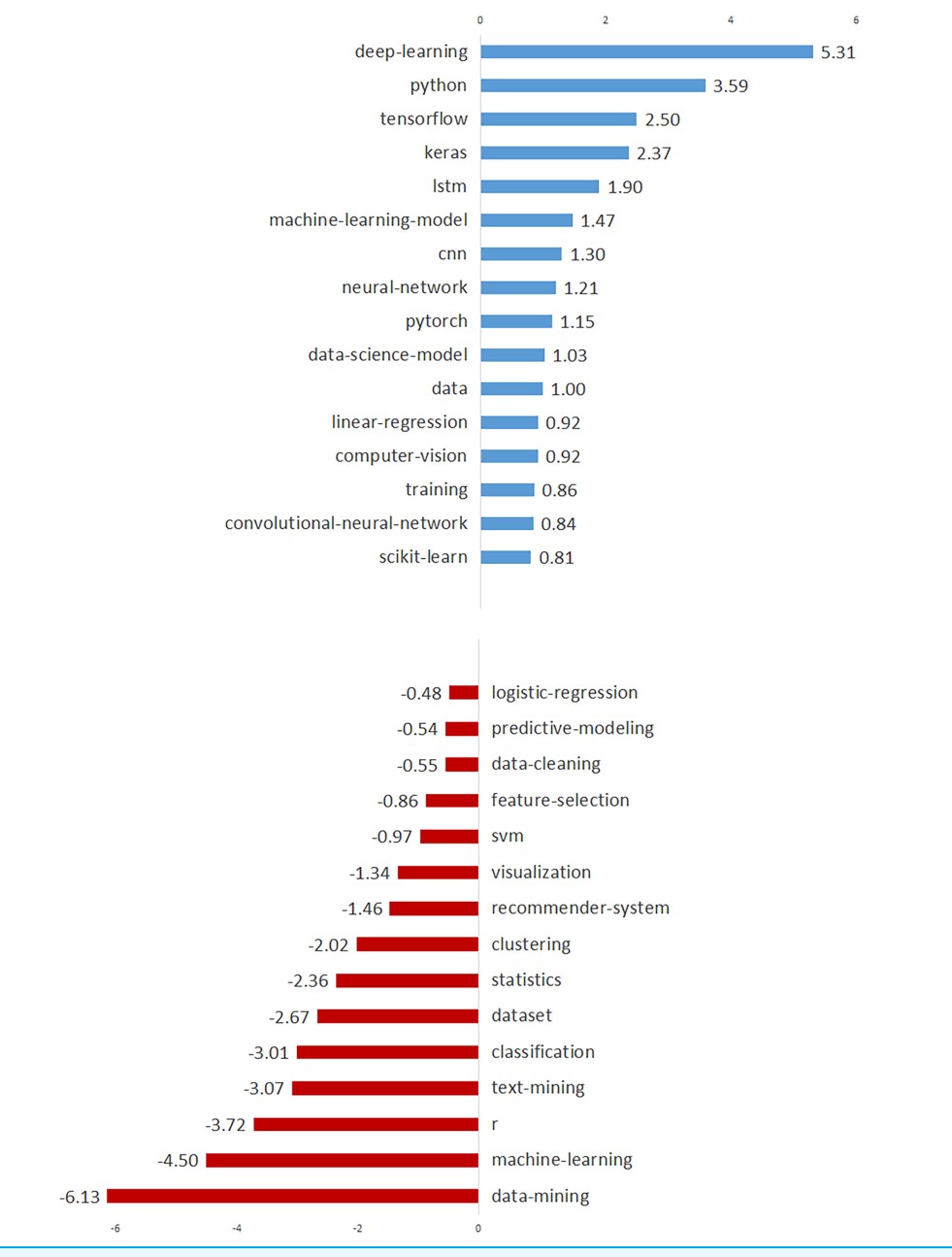

**Figure 7** The top 15 data-driven technologies with increasing and decreasing trends.

vast majority of these issues and challenges are related to how data-driven task techniques and tools are used in data science. In order to provide an understanding of the tasks, techniques, and tools used in data science, we categorized the tags based on their functions and contexts. In this way, we classified tags into three categories: tasks, algorithms, and tools.

Table 5 shows the top 25 tags for each category. The tasks, algorithms, and tools listed in Table 5 can be defined as the three pillars of data science. As shown in Table 5, the top five data science tasks are "machine-learning", "deep-learning", "classification", "nlp", and

**Table 5  Top 25 most commonly used tasks, algorithms, and tools in data science.**

| Tasks | % | Algorithms | % | Tools | % |
|---|---|---|---|---|---|
| Machine-learning | 12.28 | Neural-network | 4.85 | Python | 7.28 |
| Deep-learning | 5.28 | CNN | 2.32 | Keras | 3.11 |
| Classification | 3.57 | LSTM | 1.32 | Scikit-learn | 2.48 |
| NLP | 2.78 | Random-forest | 0.93 | TensorFlow | 2.46 |
| Time-series | 1.97 | RNN | 0.84 | R | 1.67 |
| Regression | 1.68 | Decision-trees | 0.82 | Pandas | 1.41 |
| Clustering | 1.53 | Linear-regression | 0.80 | PyTorch | 0.68 |
| Data-mining | 1.34 | Logistic-regression | 0.75 | NumPy | 0.40 |
| Predictive-modeling | 1.31 | XGBoost | 0.73 | Word2vec | 0.39 |
| Statistics | 1.21 | SVM | 0.63 | Dataframe | 0.37 |
| Feature-selection | 1.05 | Loss-function | 0.59 | Apache-spark | 0.27 |
| Image-classification | 0.85 | Word-embeddings | 0.52 | Orange | 0.20 |
| Data-cleaning | 0.81 | K-means | 0.50 | Python-3.x | 0.20 |
| Training | 0.71 | Gradient-descent | 0.47 | Matplotlib | 0.19 |
| Visualization | 0.69 | Autoencoder | 0.38 | Apache-hadoop | 0.11 |
| Cross-validation | 0.68 | PCA | 0.38 | MatLab | 0.09 |
| Reinforcement-learning | 0.68 | BERT | 0.35 | Gensim | 0.07 |
| Feature-engineering | 0.68 | Backpropagation | 0.33 | Kaggle | 0.04 |
| Computer-vision | 0.67 | GAN | 0.26 | GridsearchCV | 0.03 |
| Text-mining | 0.65 | Naive-Bayes-classifier | 0.23 | Grid-search | 0.03 |
| Class-imbalance | 0.58 | Graphs | 0.22 | Jupyter | 0.03 |
| Multiclass-classification | 0.58 | K-NN | 0.15 | VGG16 | 0.03 |
| Preprocessing | 0.56 | YOLO | 0.03 | OpenCV | 0.02 |
| Optimization | 0.52 | LDA | 0.03 | SQL | 0.02 |
| Recommender-system | 0.51 | SMOTE | 0.02 | Orange3 | 0.02 |

"time-series". The dominance of "machine-learning" and "deep-learning" tasks in data science is especially noteworthy. The most commonly used algorithms in data science are identified as "neural-network", "cnn", "lstm", "random-forest", and "rnn". The top five data science tools are discovered to be "python", "keras", "scikit-learn", "tensorflow", and "r".

## How do data science topics relate to data-driven technologies? (RQ5)
### Motivation

Earlier stages of our analysis revealed that the issues discussed by data scientists cover a wide range of tasks, techniques, and tools used in data science. Analyzing the connections between data-driven technologies (tasks, techniques, and tools) and data science issues can lead to significant discoveries. In this way, the findings of such an analysis will contribute to a better understanding of data science issues and the advancement of data science. To achieve this, we expanded our analysis at this point to investigate correlations between data science issues and data-driven technologies.

**Table 6 Topics and related data-driven technologies.**

| Topic name | Related tags |
|---|---|
| Model training | Machine-learning deep-learning neural-network keras python cross-validation tensorflow scikit-learn classification training CNN overfitting accuracy hyperparameter-tuning random-forest |
| Machine learning | Machine-learning deep-learning python neural-network classification data-mining NLP dataset data statistics bigdata predictive-modeling scikit-learn clustering algorithms |
| Neural networks | Neural-network deep-learning machine-learning keras CNN tensorflow python LSTM convolutional-neural-network RNN convolution PyTorch classification backpropagation regression |
| NLP | NLP machine-learning python text-mining word-embeddings deep-learning word2vec BERT classification neural-network named-entity-recognition sentiment-analysis text-classification NLTK data-mining |
| Time series | Time-series machine-learning LSTM python deep-learning forecasting predictive-modeling keras neural-network RNN regression classification anomaly-detection prediction tensorflow |
| Data manipulation | Python pandas machine-learning R dataframe data-cleaning dataset scikit-learn data data-mining NumPy clustering classification visualization time-series |
| Feature engineering | Machine-learning python scikit-learn feature-selection categorical-data feature-engineering regression classification correlation preprocessing feature-scaling data-cleaning neural-network R clustering |
| Regression models | Machine-learning decision-trees scikit-learn python random-forest regression feature-selection classification XGBoost linear-regression logistic-regression R predictive-modeling neural-network deep-learning |
| Statistics | Machine-learning statistics python classification regression neural-network deep-learning clustering probability data-mining dataset distribution scikit-learn R outlier |
| Tools | Python machine-learning tensorflow R deep-learning bigdata GPU dataset apache-spark keras orange scikit-learn neural-network data apache-hadoop |
| Computer vision | Deep-learning machine-learning CNN image-classification neural-network computer-vision python tensorflow keras object-detection image-recognition convolutional-neural-network classification GAN dataset |
| Classification | Classification machine-learning class-imbalance python multiclass-classification scikit-learn deep-learning neural-network multilabel-classification keras dataset naive-Bayes-classifier NLP SVM tensorflow |
| Recommendation systems | Machine-learning recommender-system python predictive-modeling classification data-mining clustering statistics dataset regression R NLP deep-learning time-series machine-learning-model |
| Loss functions | Machine-learning neural-network gradient-descent deep-learning loss-function optimization backpropagation python classification linear-regression logistic-regression mathematics regression SVM tensorflow |
| Evaluation | Machine-learning python classification visualization scikit-learn evaluation R matplotlib confusion-matrix metric regression deep-learning plotting seaborn keras |
| Clustering | Clustering machine-learning reinforcement-learning python K-means deep-learning Q-learning graphs neural-network scikit-learn unsupervised-learning DQN data-mining classification DBSCAN |
| Coding errors | Python keras tensorflow machine-learning deep-learning scikit-learn neural-network NumPy LSTM CNN PyTorch pandas NLP classification R |
| Dimensionality reduction | Machine-learning dimensionality-reduction deep-learning PCA neural-network NLP python SVM autoencoder clustering similarity classification transformer scikit-learn word-embeddings |

## Approach

At this stage, we added another process to our current analysis and attempted to correlate our findings in RQ1 and RQ3. We began by calculating the tag distribution for posts assigned to each topic discovered in RQ1. In RQ3, we explained how the tags for each post are parsed. We then identified the top 15 tags with the highest frequency for each topic. In this way, we determined which tasks, techniques, and tools are closely related to which data science issues.
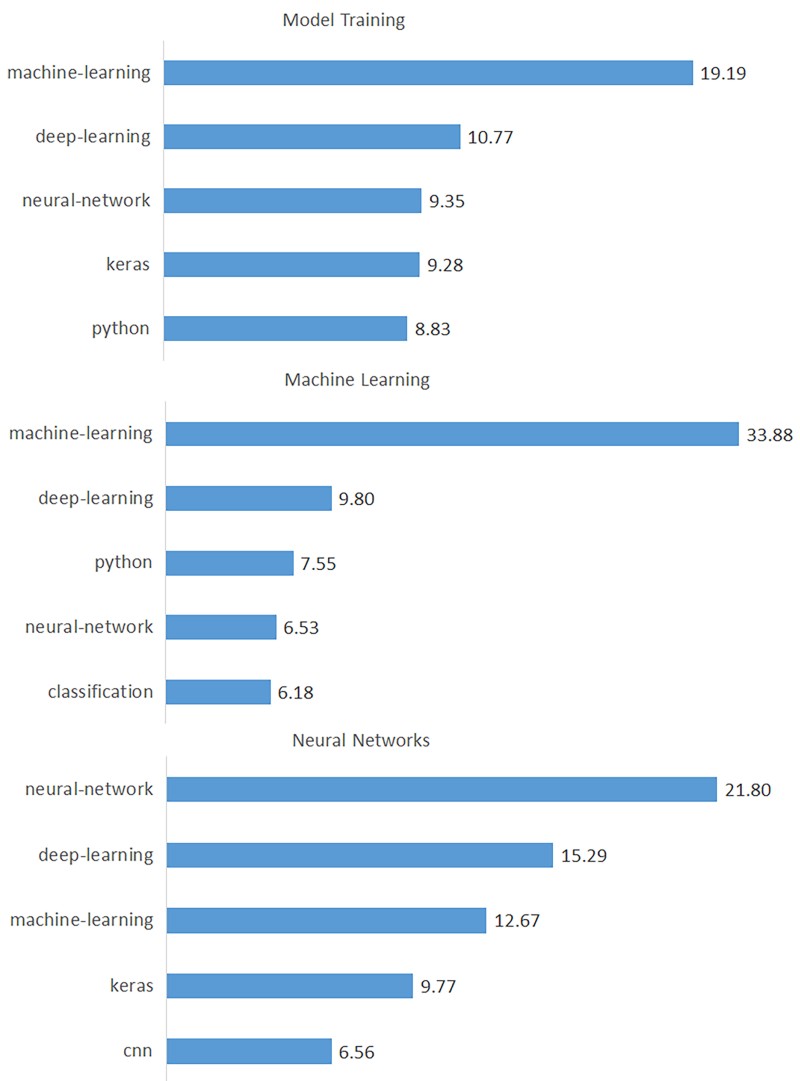

**Figure 8 Percentages of data-driven technologies for the top three topics.**

*Finding*

As a result of this process, we identified the top 15 tags for each topic and presented them in Table 6, where the topics are listed in descending order by percentage. Likewise, the top 15 tags for each topic are sorted in descending order. As shown in Table 6, the tag "machine-learning" has the highest frequency for the topic "Model Training", while "random-forest" has the lowest. Figure 8 shows the first three topics and their tags to illustrate the significance of these tags for each topic. In this way, we discovered a wide range of data-driven technologies (*e.g.*, tasks, algorithms, analysis tools, programming languages, and machine learning libraries) categorized under the 18 issues. According to the results in Table 6, "machine-learning" is the first tag in nine of the 18 topics. In other words, machine learning is clearly a dominant concept in data science issues. Python was seen as the first tag in three of these topics ("Data Manipulation", "Tools", and "Coding

Errors"). This finding highlighted Python as the most popular programming language for data science. Apart from these, the tags "neural-network", "nlp", "time-series", "deep-learning", "classification", and "clustering" appeared as the first tag in only one topic.

## DISCUSSION

Data science is comprised of dynamic and competitive working environments in which tasks, paradigms, tools, technologies, skills, and experiences are constantly updating and progressing. Our analysis identifies the most frequently asked questions by data scientists on the DSSE platform and investigates the dimensions that indicate their importance. Our findings revealed the data science issues and challenges as 18 distinct topics discovered by LDA. The findings of our study have important implications for understanding data scientists' thoughts, interactions, work flows, and issues in today's IT industry. We will now go over these findings in detail. Initially, we found that the topic "Machine Learning" and its related processes are emerging as the most dominant tasks and most common implementations for data scientists, so tools and applications support for machine learning may need to mature further (*Alshangiti et al., 2019*; *Hin, 2020*). Other machine learning-related topics with high percentages, such as "Model Training", "Neural Networks", and "Feature Engineering", support our finding (see Table 1). We frequently encountered machine learning-related keywords and tags in other topics as well. Table 6 shows that "machine-learning" is the first tag in nine of 18 topics. Furthermore, "NLP", "Computer Vision", and "Recommendation Systems" were identified to be prominent data science application areas (see Table 1) (*Hin, 2020*; *Karbasian & Johri, 2020*).

We also analyzed the temporal evolution of data science issues and discovered a number of noteworthy findings. "Model Training" is once again at the top of the list of topics with the fastest growing trend. Following that, the data science topics with the highest increasing trend were identified as "Neural Networks", "Regression Models", and "Time Series" (see Fig. 3). Furthermore, within application areas, "Computer Vision" is on the rise, while "NLP" and "Recommendation Systems" are on the decline (*Liu et al., 2017*). The strong increasing trend of "Model Training", "Neural Networks", and "Computer Vision" topics suggests that deep learning will gain a more leading position in the near future (*Karbasian & Johri, 2020*). These findings point to a significant shift from machine learning to deep learning (see Fig. 3) (*Hin, 2020*).

We expanded our findings by including some indicators of the issues in our analysis to better understand the most important, popular, and difficult issues for data scientists. The number of questions, views, and answers for a topic on the DSSE platform reveal various insights about it (*Bagherzadeh & Khatchadourian, 2019*). If a question has already been asked, it will not be asked again; instead, the user will see the previously asked question and its related answers. As a result, the number of views on the topics is an important indicator of the topic's popularity (*Uddin et al., 2021*). We discovered that the top three most popular topics were "Data Manipulation", "Coding Errors", and "Tools" (see Fig. 4). These three topics are the most widely discussed in data science. "Time Series" is the most difficult topic, with the lowest percentage of accepted answers, followed by "Computer Vision", "Recommendation Systems", "NLP", and "Statistics" (see Fig. 5) (*Sarker, 2021*).

Such common issues in data science that have yet to be resolved should be investigated further and supported by data scientists (*Cao, 2017*).

Our study also found that data science issues and challenges are closely related to data-driven technologies (tasks, techniques, and tools) (*Cao, 2017*; *Sarker, 2021*). The most frequently used data-driven technologies are "machine-learning", "python", "deep-learning", "neural-network", and "classification" (see Table 4 and Fig. 6) (*Hin, 2020*). The prominent tags with an increasing trend are "deep-learning", "python", "tensorflow", "keras", and "lstm" (see Fig. 7) (*Karbasian & Johri, 2020*). Deep learning will be the most focused area in data science in the near future, according to the top five tags with the highest increasing trend (*Karbasian & Johri, 2020*). On the other hand, the top five tags with decreasing trend were identified as "data-mining", "machine-learning", "r", "text-mining", and "classification" (see Fig. 7).

Another significant finding is that, while the trend of "data mining" and "machine learning" tags is decreasing, deep learning-related tags such as "deep-learning", "tensorflow", "keras", and "lstm" are increasing (see Fig. 7). This discovery highlights a significant shift in data science from "machine learning" to "deep learning" (*Hin, 2020*; *Karbasian & Johri, 2020*). Based on our findings, we can also speak of a significant shift from "R" to "Python", which are data-oriented programming languages (see Fig. 7). Furthermore, among the important findings of our study are the most commonly used tasks (*i.e.*, "machine-learning", "deep-learning", and "classification"), algorithms (*i.e.*, "neural-network", "cnn", and "lstm") and tools (*i.e.*, "python", "keras", and "scikit-learn") in data science (see Table 5) (*Cao, 2017*; *Karbasian & Johri, 2020*).

## Implications of findings

Knowledge and experiences shared on Q&A platforms like DSSE should motivate researchers, practitioners, and developers to create documentation aimed at resolving common data science issues. We have inferred remarkable implications and guidelines for data science stakeholders based on the empirical background, methodology, and findings of this study, which will contribute to their understanding of the field. We hope that our implications will help data science communities with various profiles, such as developers, researchers, practitioners, educators, and enthusiasts. Developers can lead data science innovation by creating more specific tools and applications to address the current issues and needs of data scientists, which are also highlighted in our findings. One of the most popular topics is "Data Manipulation". For such widespread issues, tool developers can create useful libraries or tools. Developers can use our findings to improve data-driven tools or to choose which frameworks and libraries to support.

Identifying the questions that data scientists are asking on platforms like DSSE can help the research community better understand the challenges of data science. While all of the issues identified are important in their own right, our findings indicate that data science researchers should prioritize the most visible and difficult issues. Researchers can also use our methodology for experimental research and analysis in a variety of settings. Furthermore, data science educators can create more tutorials to assist in the training of data scientists and candidates, especially considering the difficulty of the topics as well as

the most commonly used tasks, algorithms, and tools. Educators can keep their training programs and curricula up to date with current field trends, allowing them to provide up-to-date background to data scientists training. Our findings can be used by DSSE and other Q&A platforms to develop new approaches for contextually tagging posts and better categorizing user posts. Data science enthusiasts and general readers may find our findings useful in keeping up with emerging developments and trends in the data science industry and ecosystems. More researchers in this field can also contribute to the improvement of data science processes. We hope that our findings, which highlight the challenges that data scientists face, will help to guide future research in this area.

## CONCLUSIONS

This research aims to shed light on common issues and challenges encountering data scientists. To that end, all posts shared on the DSSE platform were analyzed using LDA-based semantic topic modeling. Furthermore, the most commonly used data-driven technologies and their connections to data science issues were investigated. Our research methodology is based on the adaptation and implementation of LDA, an unsupervised generative approach for semantic topic modeling that is widely used in textual content analysis.

As a result of the LDA analysis, 18 topics were identified that demonstrate the current landscape of data science issues and trends. Among these topics, "Model Training", "Machine Learning", and "Neural Networks" were the most frequently asked. Furthermore, the most viewed (most popular) topics were "Data Manipulation", "Coding Errors", and "Tools". The most difficult topics were identified as "Time Series", "Computer Vision", and "Recommendation Systems".

One of our key discoveries is that data science issues and challenges are inextricably linked to data-driven technologies (tasks, techniques, and tools). While the trend for "data-mining" and "machine-learning" tags is decreasing, the trend for deep learning-related tags such as "deep-learning", "tensorflow", "keras", and "lstm" is increasing. As the findings show, there has been a significant shift in data science from machine learning to deep learning. It was also determined that the most commonly used algorithms are "neural-network", "cnn", "lstm", "random-forest", and "rnn", and the most commonly used tools are "python", "keras", "scikit-learn", "tensorflow", and "r". Thus, by analyzing the issues discussed by data scientists, this study provides an in-depth understanding of this dynamic discipline.

These findings will help online communities with diverse profiles understand data science focuses and issues. Our findings have significant implications for the various data science stakeholders who are working to advance data science. Our findings can be used by tool builders to improve support and documentation, by developers to create data applications and libraries, and by educators to create modern training and curricula. By focusing on current issues, researchers can provide more solutions for data science. Our methodology can also be applied to other developer platforms like forums, blogs, and portals, as well as different Q&A platforms like Kuggle, Reddit, and Quora.

### Funding
The author received no funding for this work.

### Competing Interests
The author declares that they have no competing interests.

### Author Contributions
- Fatih Gurcan conceived and designed the experiments, performed the experiments, analyzed the data, performed the computation work, prepared figures and/or tables, authored or reviewed drafts of the article, and approved the final draft.

### Data Availability
The raw data and codes are available in the Supplemental File.

### Supplemental Information
Supplemental information for this article can be found online at http://dx.doi.org/10.7717/peerj-cs.1361#supplemental-information.

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
