# Peer review of "What issues are data scientists talking about? Identification of current data science issues using semantic content analysis of Q&A communities"

_PeerJ Computer Science, doi:10.7717/peerj-cs.1361_

## Round 0.1 · original submission · Major Revisions

Please read the reviewers' comments carefully and revise your manuscript accordingly.

Reviewer 1 ·

Basic reporting

The author employs LDA to perform topic modeling on Stack Exchange data related to Data Science and draws statistical conclusions on the popular topics/tools/trends in the field, which may be of interest for people working in the field of Data Science.
The paper is written in professional english, well-structured and easy to follow. The methodology is clearly described and results are valid and well presented.
The only concern is that the dataset used in this paper as well as the methodology (including the data preprocessing step and the LDA approach) is very similar to the work done by Karbasian & Johri in 2020, which the author also cited in the paper, making the paper less innovative. However, given the fact that this paper covered more findings compared to (Karbasian & Johri, 2020), as the author claimed, expanding the scope of the previous paper, this concern should be a minor one.

Experimental design

The methodologies used in the paper, including data preprocessing, LDA algorithm, as well as other statistical methods are rigorous and well designed, leading to solid results.

Validity of the findings

Findings are valid and statistically sound; Figures and tables are well presented and easy to follow for readers.

Reviewer 2 ·

Basic reporting

The author uses latent Dirichlet allocation (LDA) to learn topics embedded in data science discussions, where the main source of the data is from DSSE, a data science-focused Q&A website. And the author further try to answer 5 questions:
1. What topics are discussed by data scientists?
2. How do the data science topics evolve over time?
3. How do the popularity and difficulty of the topics vary?
4. What are the most commonly used tasks, techniques, and tools in data science?
5. How do data science topics relate to data-driven technologies?

Experimental design

no comment

Validity of the findings

This paper discusses some important, popular, and difficult issues for data scientists are facing currently. However, the implication of these findings are not clear, and all the topics that are shown in the paper are already hot topics, such as NLP, computer vision, etc. Each year, thousands of articles are published in those areas each year. Would the findings in this paper truly have significant implications on the challenges of data science? I think the results and indications are not giving significant implications.

Additional comments

It is great to do such analysis, but the results don't really reveal the pain points that data scientists care about. I think the topics need to be more specific, unlike general Machine learning and Deep learning. Say, for NLP, the author can use Text Summarization to analyze the current issue in NLP area, this would be more insightful.

Reviewer 3 ·

Basic reporting

With the development of massive amounts of information and data science, data scientists often use specific question-and-answer websites to find solutions to difficult problems. In the paper, the author uses the LDA method to to explore five research questions about data science on the DSSE website. However, for the important method LDA, there is no formula or flow chart in the article to introduce it in detail. I think using formulas or a flow chart would make it clearer to readers know how the LDA works in the article.

Experimental design

The author design RQ2 to explore how do the data science topics evolve over time? However, the results of RQ2 presented in Table3, Figure4 and Figure5 do not show the changing trend of data science topics over time. And there are some errors in the finding of RQ2, because it is the same as the finding of RQ3.

Validity of the findings

The research results of the article are helpful for data workers to better understand and use data science. However, I am very concerned that the significance is limited, and perhaps the author can use some examples to strengthen the significance.

Additional comments

There are some repeated paragraphs in the article. For example, the finding part of RQ2 and RQ3 mentioned in the previous comments is same. Besides, there are some grammatical errors. I hope the author can check the content of the article more carefully.

---

## Round 0.2 · accepted · Accept

Considering all comments from three reviewers and the major revision undertaken, I would like to recommend publication in PeerJ.

Reviewer 2 ·

Basic reporting

The author uses the LDA method to explore five research questions about data science on the DSSE website. I can see some questions are addressed, but I think the paper is not novel, although the article is somewhat educational for some data workers. However, I am very concerned that the significance is limited, and perhaps the author can put this on some other learning website. Overall, I am missing somewhat interesting turns in terms of methods or even surprises, which limits my enthusiasm to a certain degree.

Experimental design

No comment.

Validity of the findings

No comment